# Spatial Configuration and Extent Explains the Urban Heat Mitigation Potential due to Green Spaces: Analysis over Addis Ababa, Ethiopia

**Berhanu Keno Terfa [1]** , **Nengcheng Chen [1,2]** , **Xiang Zhang [1,***] and Dev Niyogi [3,4]**

[1]   State Key Laboratory of Information Engineering in Surveying, Mapping, and Remote Sensing,
     Wuhan University, Wuhan 430079, China; berekeno@whu.edu.cn (B.K.T.); cnc@whu.edu.cn (N.C.)
[2]   Collaborative Innovation Center of Geospatial Technology, Wuhan 430079, China
[3]   Department of Geological Sciences, Jackson School of Geosciences, University of Texas at Austin,
     Austin, TX 78712, USA; happy1@utexas.edu
[4]   Department of Civil, Architectural, and Environmental Engineering, Cockrell School of Engineering,
     University of Texas at Austin, Austin, TX 78712, USA
*   Correspondence: zhangxiangsw@whu.edu.cn; Tel.: +86-158-7147-2170

**Abstract:** Urban green space (UGS) is considered a mitigative intervention for urban heat. While increasing the UGS coverage is expected to reduce the urban heat, studies on the effects of UGS configuration have produced inconsistent results. To investigate this inconsistency further, this study conducted a multi-spatial and multi-temporal resolution analysis in the Addis Ababa city metropolitan area for assessing the relationship between UGS patterns and land surface temperature (LST). Landsat images were used to generate land cover and LST maps. Regression models were developed to investigate whether controlling for the proportion of the green area (PGS), fragmentation, shape, complexity, and proximity distance can affect surface temperature. Results indicated that the UGS patches with aggregated, regular and simple shapes and connectivity throughout the urban landscape were more effective in decreasing the LST as compared to the fragmented and complicated spatial patterns. This finding highlighted that in addition to increasing the amount of UGS, optimizing the spatial structure of UGS, could be an effective and useful action to mitigate the urban heat island (UHI) impacts. Changing the spatial size had a significant influence on the interconnection between LST and UGS patterns as well. It also noted that the spatial arrangement of UGS was more sensitive to spatial scales than that of its composition. The relationship between the spatial configuration of UGS and LST could be changed when applying different statistical methods. This result underlined the importance of controlling the effects of the share of green spaces when calculating the impacts of the spatial configuration of UGS on LST. Furthermore, the study highlighted that applying different statistical approaches, spatial scale, and coverage of UGS can help determine the effectiveness of the association between LST and UGS patterns. These outcomes provided new insights regarding the inconsistent findings from earlier studies, which might be a result of the different approaches considered. Indeed, these findings are expected to be of help more broadly for city planning and urban heat mitigation.

**Keywords:** urban green space; land surface temperature; spatial patterns; urbanization; Addis Ababa

## 1. Introduction

Over half the global population dwells in cities [1]. Urbanization is typically associated with providing better socio-economic conditions. On the other end, urbanization has also been associated with environmental and ecological stresses [2–4]. One of the widely recorded and evident impacts

associated with urbanization is the increased temperature over the cities relative to the adjacent rural areas, or the urban heat island (UHI) [5,6]. Relative to the rural regions, urban areas have distinct morphology, geometry, and infrastructure that contributes to the UHI. Changes in land use and land cover result in an abundance of built impervious surfaces and depletion of green vegetation covers, causing alterations in the surface energy fluxes [6–9], distribution and composition of biodiversity [10,11], increasing anthropogenic heat releases [12], and human health [13,14]. As a result, how to understand and mitigate these adverse impacts has become a key study topic in urban ecology, geography, and climatology [15–18].

Spatial patterns and intensity of UHI are determined by the size of the population and impervious surface [4,19–21], land cover types [15,22–25], spatial patterns of green space [16,26–28], urban spatial structure [2,4,29–32], socioeconomic activities [33,34], and climate conditions [15,35–37]. Broadly, two major categories of UHI have been identified depending upon the substrate of their influences: air (atmospheric) and surface. The atmospheric UHI and the surface temperature are highly correlated to health, though the air UHI impact is reflected to be intimately connected to human health [38]. The in-situ measurement techniques used for atmospheric UHI monitoring [37–40], have the advantages of better temporal resolution but lack spatial resolution. Whereas the surface land surface temperature (LST) is typically mapped using satellite data from thermal sensors [41–43]. Thermal remote sensing makes use of non-contact instruments to sense thermal radiations and estimate surface or skin temperatures. Nevertheless, thermal remote sensing has its limitation of poor temporal coverage, coarse spatial resolution, and the requirement for clear weather conditions.

Many studies have estimated LST from medium-resolution remotely sensed satellite data (e.g., Landsat) for long-term temperature observations from the early 1980s. For instance, the significant cooling effects of urban green space (UGS) spatial patterns on the urban heat have been widely demonstrated by previous researches [22,24,35,44–47]. Spatial pattern encompasses composition (the amounts and diversities of land cover types), as well as configuration (their distribution and structure). Urban landscapes are composed of several diverse land cover varieties, while built-up and vegetated cover are two essential features in the UHI formation [48]. Accordingly, it is known that increasing the amount of green space can significantly reduce the ambient air and land surface temperature [16,43,44]. This correlation has been reported consistently and follows the first principle surface radiative flux balance at the surface. Several studies also confirmed that the arrangement (configuration) of green space had considerable impacts on land surface temperature [2,26,31,49,50]. However, these studies produced contradictory findings. Green space with higher patch density, for instance, decreased LST in research carried out in the Phoenix metropolitan area, USA [51], Lagos, Nigeria [39], Shanghai, China [16], regions across Illinois-Indiana-Ohio, USA [19], and Berlin, Germany [52]. On the other hand, green space density was associated with increased LST in Jakarta, Indonesia [53], and Nairobi, Kenya [54]. In addition, the edge density of green space had negatively correlated to LST with several cities [42,43,49,51], and contrarily, it was positively correlated with others [45,55]. These inconsistencies are likely due to the following reasons. (i) using a single snapshot data for individual cities with multiple approaches [39,45]; (ii) implementing different images of data with varied spatial resolutions [21,42,55,56]; (iii) applying a different statistical analysis [2,16,45,53]; and (iv) using a different spatial scale (analytical unit) such as circular plots [26,51,57,58], sub-zones (16,30,43,45), pixels or grids [59], or city blocks [52].

Although the above-stated factors could be reasons for the inconsistency findings in the prior studies, such contradictory results could restrict the application of findings for planning and managing the urban green space. Hence, further investigation is warranted. Moreover, urban regions are complex systems [20,60]. The phenomenon associated with urban landscape characteristics is scale-dependent. When the scale of observation is altered, the correlation between pattern and process phenomenon will change [16,42,45]. Hence, recognizing how the urban green space pattern influences the surface energy balance and the variability of LST needs, the analytical size exemplifies the characteristic spatial extent of LST and green space interaction. As noted in previous studies, there are two broad

methods to accomplish this aim: (1) using a single city study to explore temporal dynamics [43,57], and (2) a comparative study of multiple cities applying similar approaches [45,53]. The present study adopts the first approach, to address the following questions: (i) does the spatial pattern of UGS affect temperatures differently in the city with changes over time? and (ii) do different statistical approaches or analytical units result in different effects of the spatial configuration of UGS on LST?

To answer these questions, the study analyzed the relationships between the LST and spatial configuration of UGS by using different statistical methods, and varied spatial configuration across different periods in the metropolitan area of Addis Ababa, Ethiopia. Landsat images were used to produce the land cover and LST maps. As cities have mostly limited space for greening [16,29,43], this study provides empirical results that can improve the understanding of the seemingly contradictory impacts of the spatial configuration of UGS patterns on LST noted in earlier studies. Such empirical results can potentially help planners and managers in regulating the spatial arrangement of urban green space to mitigate UHI. Furthermore, the adopted integrated approaches, multiple statistical, and landscape metrics analysis at varied analytical units, provide comprehensive and detailed information for environmental planning issues.

## 2. Materials and Methods

### 2.1. Study Area

The study area, presented in Figure 1, includes the city of Addis Ababa and its surrounding regions, which cover 1711.74 km$^2$. It is located between 8°54′00″ N to 9°8′00″ N and 38°36′00″ E to 38°54′00″ E. The mean altitude of the study site is 2685 m above sea level. Addis Ababa is the capital, and most important commercial, political, and industrial city of Ethiopia. The city also serves as the seat of the African Union in addition to the Oromia National, Regional State. The city has a total area of about 527 km$^2$ and 4.3 million residents in the year 2017 [3], which cover 16.9 percent of the country's urban inhabitants and 3 percent of the country's overall inhabitants. The city's population is expected to grow to 5.9 million in 2030 at a mean annual growth rate of 4.1 percent [61]. The climate in the study site is warm and temperate. Typically, March, April, and May are the warmest months of the year, with values of 25 °C for the mean highest temperatures. November and December are the coldest months with the lowest average temperature values of 8 °C. The rainfall is moderate to heavy during the summer (June to August) with a mean yearly rainfall of 1143 mm. The dispersed nature of Addis Ababa city's spatial growth [3] and the rapid reduction of urban green space [62,63] are likely to have considerable impacts on spatial patterns of temperature [42,51,64]. Hence, Addis Ababa metropolitan area was an interesting case study to explore the environmental effects of urbanization.

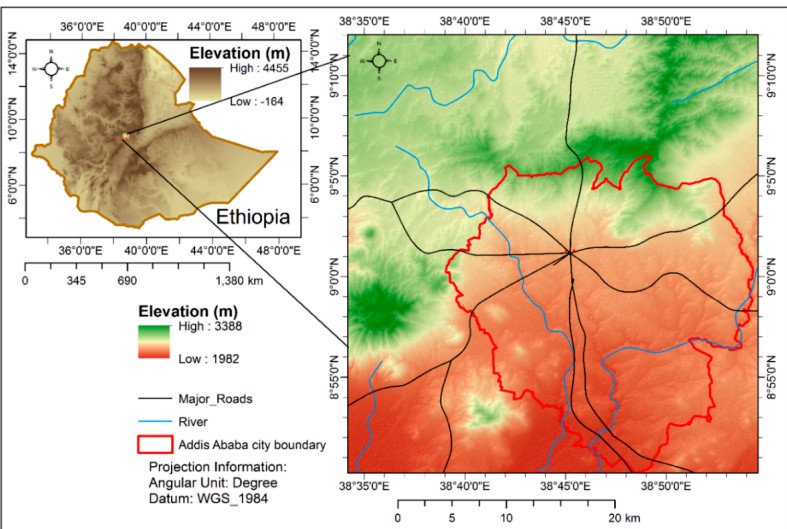

**Figure 1.** Map of the study area.

## 2.2. Remote Sensing Data and Processing

Landsat is regularly adopted for computing LST in UHI investigations [43] and is openly available. Three Landsat images for 1987, 1999, and 2019 acquired during the dry season (February) with clear atmospheric conditions were used in this study. These images were chosen based on the availability of the data for the same months. This month (February) was selected based on previous studies that reported the UHI effects during the dry season were more significant [34,59]. The images were downloaded from the United States Geological Survey. The details of the data implemented for the study are shown in Table 1. The multispectral bands were applied for classification of land-use/land cover (LULC) classes, while the thermal bands were used to calculate satellite brightness temperatures.

**Table 1.** Remotely sensed data information for the study sites.

| Satellite | Sensor | Path/Row | Acquisition Date | Season | Source |
|---|---|---|---|---|---|
| Landsat-5 | TM | | 1987/02/09 | | |
| Landsat-5 | TM | 168/54 | 1999/02/10 | Dry | www.earthexplorer.usgs.gov |
| Landsat-8 | OLI/TIRS | | 2019/02/01 | | |
| Digital elevation model (DEM) data (30-m spatial resolution) | | | | | |

## 2.3. Land Cover Mapping

The general idea of remotely sensed data and the methodological method employed for this investigation are illustrated in Figure 2 flow diagram.

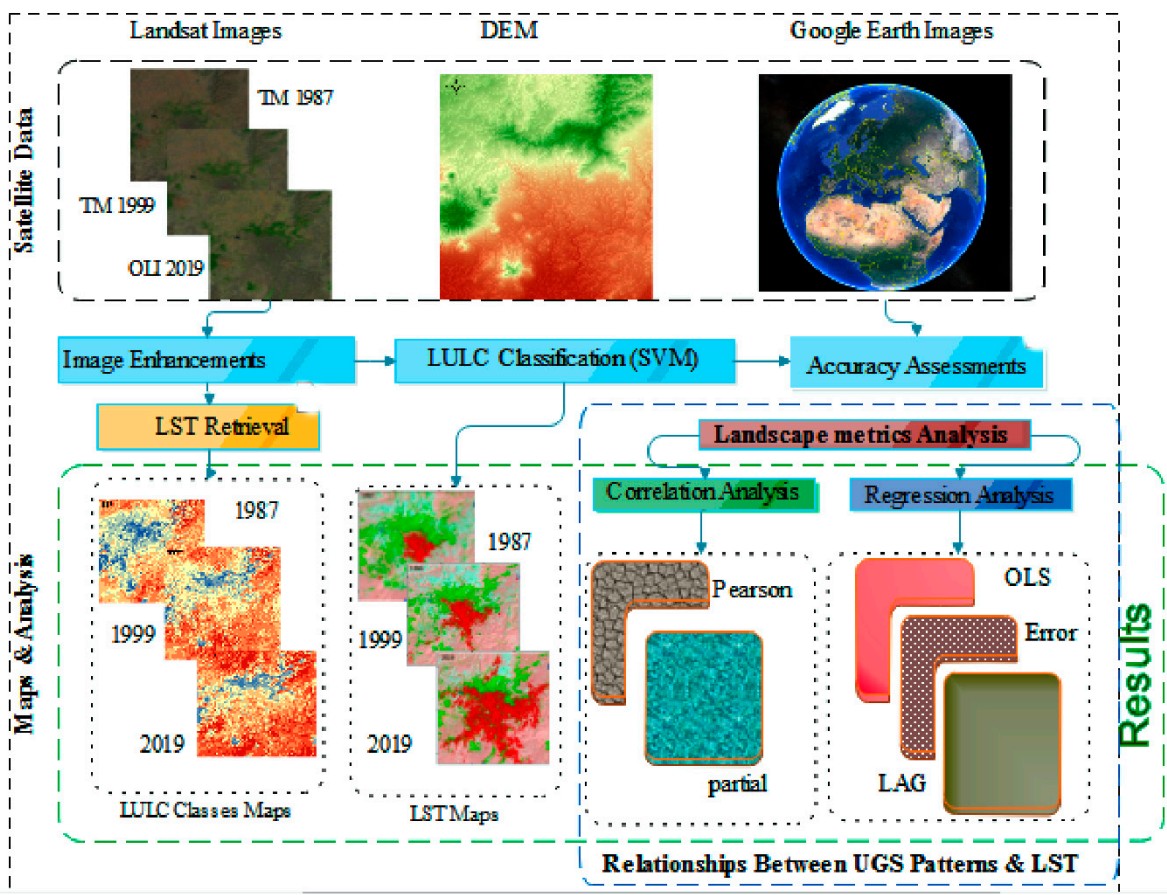

**Figure 2.** The general workflow for the relationship between UGS patterns and LST analysis in the study.

Satellite data pre-processing, spectral, spatial, and radiometric rectifications were done using standard remote sensing methods [65–67]. The study then mapped the LULC classes based on 30 m resolution of Landsat images, bands 5-4-3 for Landsat 8 OLI, and bands 4-3-2 for Landsat-5. Depending on its spectral signature, each pixel in the image was allocated to the category that fits best. Ancillary data, such as Google Earth images, were used to help in classification. The five defined categories were: built-up area, bare land, water body, vegetation cover, and grassland. A supervised classification approach with the support vector machine (SVM) classifier algorithm was employed. The correctness of land cover classified results was validated by the accuracy assessment approaches [68–70]. The study further merged the vegetation cover, grasslands, and water body as UGS to analyze the relationships between the spatial patterns of UGS and land LST.

### 2.4. Land Surface Temperature Retrieval

Several factors affect LST, including surface emissivity [71,72], elevation [19,73], wind velocity, time of day, and the phenomena of precipitation [74], clouds [4,43,72], atmospheric situations, satellite sensor features, and the mathematical approaches used to retrieve LST [75,76]. It is not feasible to explicitly account for all these components in this study due to the following reasons: (1) the atmospheric data are not available before January 2000. (2) Though the atmospheric rectification based on satellite imagery using generic parameters is important, it has been revealed to the result in only a minor error for LST retrieval [77]. (3) This study implemented LST for the exploration of UGS patterns' effect on urban land temperatures.

### 2.4.1. Retrieval of Spectral Radiance

To estimate LST, the study used thermal infrared band 6 (10.40–12.50 μm) for Landsat 5-TM and band 10 for Landsat-8 OLI (10.3–11.3 μm) with resolutions of 120 m and 100 m, respectively. As the Landsat images were produced from continuous digital values (digital numbers), to obtain LST, the digital number of pixels for each thermal image was converted to the top-of-atmosphere (TOA) radiance using the Equations (1) and (2) [4,30,64,65]. For Landsat 5-TM (TIR band 6), Equation (1) and for Landsat 8 (TIR band 10), Equation (2) were used as follows:

$$L_\lambda = \frac{L_{max} - L_{min}}{QCAL_{max} - QCAL_{min}} \times (QCAL - QCAL_{min}) + L_{min} \tag{1}$$

where:

(a)   $L_\lambda$—TOA Spectral Radiance $\left(W/\left(m^2 \times sr \times \mu m\right)\right)$;
(b)   $LMAX_\lambda$—the spectral radiance $(QCALMAX (W/(m^2 \times sr \times \mu m)))$;
(c)   $LMIN_\lambda$—the spectral radiance $\left(QCALMIN \left(W/\left(m^2 \times sr \times \mu m\right)\right)\right)$;
(d)   $LMIN_\lambda = 1.238$; $LMAX_\lambda = 15.303$;
(e)   QCAL—the quantized rectified pixel value in Digital Numbers (DN); and
(f)   QCALMAX and QCALMIN the—maximum and minimum quantized adjusted pixel values matching to $LMAX_\lambda$ in DN = 255 and $LMIN_\lambda$ in DN = 1, respectively.

$$L_\lambda = M_\lambda \times QCAL_\lambda + A_\lambda \tag{2}$$

where, $M_\lambda$ and $A_\lambda$ = Radiance multiplicative and additive scaling factor for the band, respectively, and $QCAL_\lambda$ = the quantized calibrated pixel value in DN. The study then converted the $L_\lambda$ values to at-satellite brightness temperature (TB) using (Equation (3)).

$$T_B = \frac{K_2}{\ln\left(\frac{K_1}{L_\lambda} + 1\right)} \tag{3}$$

where, TB values in Kelvin (K); K1, K2 = Thermal conversion constants for the band.

### 2.4.2. Estimation of the Land Surface Emissivity

Land surface emissivity ($\varepsilon$) is one of the essential parameters in retrieving LST. Surface materials have different emissivity due to their different viewing angles, materials, and roughness. Normalized difference vegetation index (NDVI) is one of the most commonly applied vegetation indexes. NDVI is calculated from the red (R) and near-infrared (NIR) bands—for Landsat 5, and 8, using Equation (4).

$$\text{NDVI} = \frac{\text{NIR} - \text{RED}}{\text{NIR} + \text{RED}} \tag{4}$$

where, RED is a red spectral band, and NIR is a near-infrared spectral band of Landsat images. Hence, to compute the LSE, the NDVI-based emissivity approach proposed by Van de Griend and Owe [78] was applied in this study (Equation (5)).

$$\varepsilon = \varepsilon_v P_v + \varepsilon_s (1 - P_v) + d_\varepsilon \tag{5}$$

where, $\varepsilon_s$ and $\varepsilon_v$ denote the emissivity of bare soil pixels and vegetation, respectively. Where $P_v$ indicates the fraction of vegetation [79], which was calculated using Equation (6):

$$P_v = \left[\frac{\text{NDVI} - \text{NDVI}_{\min}}{\text{NDVI}_{\max} - \text{NDVI}_{\min}}\right]^2 \tag{6}$$

In the above, $\text{NDVI}_{\min}$ and $\text{NDVI}_{\max}$ are 0.2 and 0.5, respectively in a universal condition [80]. The term $d_\varepsilon$ includes geometrical distribution influence of the internal reflections and natural surface. As suggested by Valor and Caselles [81], the values of $\varepsilon_s$ and $\varepsilon_v$ are 0.960 and 0.985, respectively, for vegetation structures and unknown emissivity. The study also applied these emissivity values in the calculation. Moreover, for rough and heterogeneous surfaces, $d_\varepsilon$ can increase to 2% of emissivity [80] and the values of $d_\varepsilon$ are calculated as proposed by Sobrino et al. [82], using Equation (7):

$$d_\varepsilon = (1 - \varepsilon_s)(1 - P_v) \times F \times \varepsilon_s \tag{7}$$

where, F is a shape factor with a mean value of 0.55, assuming different types of geometrical distributions [82]. Finally, the rectified land surface temperature maps (Figure 3) were produced using the Equation (8) as:

$$\text{LST} = \frac{T_B}{\left(1 + \lambda \left(\frac{T_B}{P}\right) \times \ln[\varepsilon]\right)} \tag{8}$$

where, LST = the Land Surface Temperature in Kelvin (K); $p = hc/k$ ($1.438 \times 10^{-2}$ mk); h = Planck constant ($6.626 \times 10^{-34}$ J s$^{-1}$), and c = velocity of light ($2.998 \times 108$ ms$^{-1}$); k = Boltzmann constant ($1.38 \times 10^{-23}$ – J K$^{-1}$), and $\lambda$ = the wavelength of emitted radiance. To obtain the LST values in Celsius (°C), 273.15 was subtracted from the original values (K). All steps were carried out by developing a raster calculator model using ArcGIS version 10.4.1.

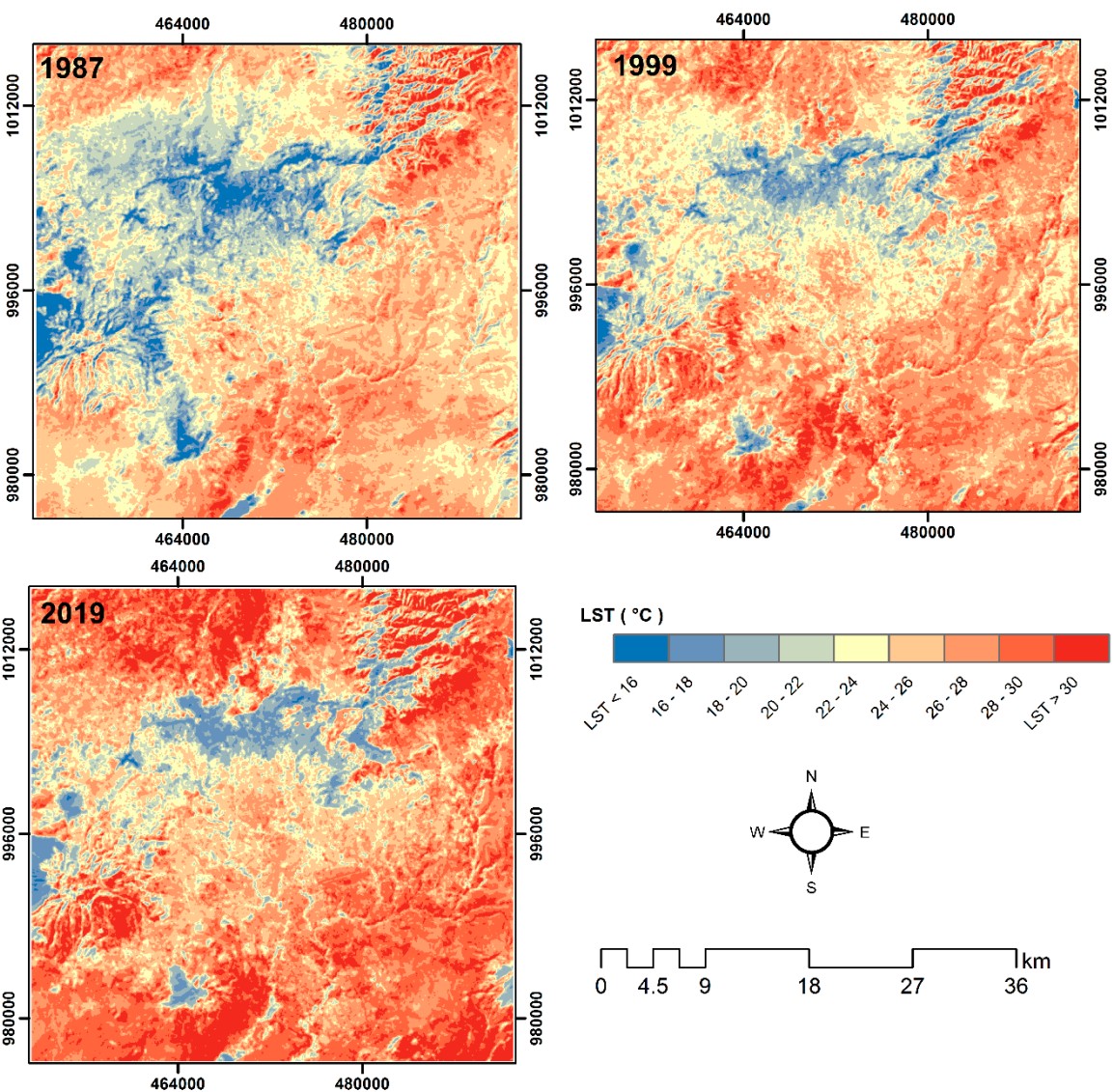

**Figure 3.** LST Maps of Addis Ababa metropolitan area from 1987 to 2019.

*2.5. The Spatial Pattern of Urban Green Space Analysis*

Four landscape metrics: (1) Percent cover of green space (PGS) (among composition metrics), and three configuration metrics: (2) patch density (PD), (3) edge density (ED), and (4) mean nearest-neighbor distance (MNN) (Table 2) were applied to compute the spatial patterns of UGS. These metrics denote the main characteristics that can describe the spatial patterns of UGS, including composition, complexity, configuration, and fragmentation of patches. These metrics were chosen based on their prevalence in the scientific literature and environmental significance [2,16,39,49,53]. The spatial metrics are sensitive to scales in the spatial pattern explorations [16,45,57]. The dependency on the scale also impacts on the results of UGS and LST analysis [16,45]. Thus, the study used 9 varied sizes of analytical units starting from 120 m (which is equal to the pixel size of the TM thermal band) to 1080 m. Accordingly, 120 m × 120 m, 240 m × 240 m, 360 m × 360 m, 480 m × 480 m, 600 m × 600 m, 720 m × 720 m, 840 m × 840 m, 960 m × 960 m, and 1080 m × 1080 m (Figure 4) were applied to determine the effects of spatial scale and distinguish the optimum spatial size in the UGS and LST relationship analysis. The study then calculated the chosen metrics and the values over generated grids using the Patch Analyst extension in ArcGIS. The LST for every analytical unit was computed using the developed-in Zonal Statistic tool of ArcMap ArcMap 10.4.1.

**Table 2.** Chosen landscape metrics implemented from McGarigal [83].

| Metric | Formula | Units | Narrative |
|---|---|---|---|
| Percent of urban green spaces (PGS) (%) | $PGS = p_i = \frac{\sum_{j=1}^{m} a_{ji}(100)}{A}$ | Percent | $p_i$ = share of the landscape occupied by patch type (class) i. $A$ = area of entire landscape (m$^2$). |
| Patch density (PD) | $PD = \frac{n_i}{A}(10,000)(100)$ | Number/100 hectares | $n_i$ = number of patches in the landscape of patch type (class) i. |
| Edge density (ED) | $ED = \frac{\sum_{k=1}^{m} e_{ik}}{A}(10,000)$ | Meter/hectare | $e_{ik}$ = whole length (m) of edge in landscape containing patch type (class) i. |
| Mean nearest-neighbor distance (MNN) | $MNN = h_{ij}$ | Hectare | $h_{ij}$ = distance (m) from patch ij to closest adjoining patch of the same type (class). |

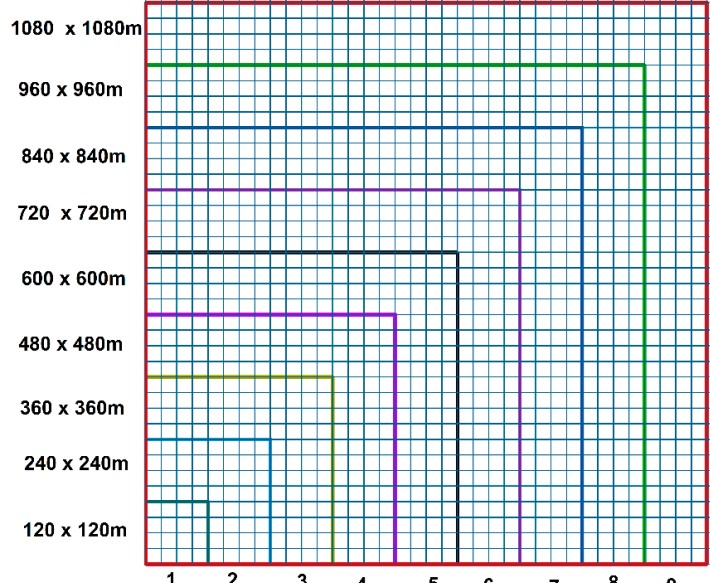

**Figure 4.** Nine analytical units (spatial scales) ranging in multiple of 120 m (Landsat image grid resolution) from 120 m × 120 m to 1080 m × 1080 m grains.

*2.6. Statistical Analysis*

2.6.1. Correlation Analysis

A Pearson correlation matrix was built based on the correlation between LST and the spatial patterns of the UGS. A partial correlation analysis was performed to assess the configuration metrics (PD, ED, and MNN) and the LST by controlling for the influence of the PGS. Controlling for the influence of PGS is important, as it had the highest correlation with the configuration metrics as well as the LST across the study period in almost all tested spatial scales. Thus, the Pearson correlation evaluation is reliable to get a spurious connection between configuration metrics and LST.

2.6.2. Regression Analysis

For each of the studied period, multiple regression models were built to select the best model that describes the highest variance in LST. To avoid redundancy (inter-correlation) within the chosen spatial metrics [43,84], a multi-collinearity condition number (MCN) coupled with the variance inflation factor (VIF), for each metric was calculated. The values of MCN and VIF for each independent variable were less than 30 and 7.5, respectively, which is acceptable for modeling [43].

The study then employed the ordinary least squares (OLS) for every built model, with the assumption of independent error terms. OLS is usually the initial step for diagnostic and to use as a benchmark for comparison. This method neglected the issue of spatial autocorrelation [45,53]. Spatial data contains spatial dependence and heterogeneity from spatial autocorrelation, which can

amplify confounding in the analysis. Spatial clustering in error may come from spatial heterogeneity or spatial dependence. To address this, spatial auto-regression (SAR) models that incorporated spatial autocorrelation into the modeling are more suitable to explore the association between spatial patterns of UGS and LST [43,45,85]. Regarding SAR, the neighborhood interconnection of the response variable was estimated by a (n × n) framework of spatial weights and was incorporated in the ordinary multiple linear regression to consider spatial autocorrelation [86]. Accordingly, two spatial regression approaches: spatial lag and error models were applied to investigate the impacts of the spatial pattern of UGS on LST. The spatial lag model expects that the spatial autoregressive occurs only in the response variable [86]. The spatial lag model is expressed in Equation (9).

$$y = \rho W y + \beta X + \varepsilon \tag{9}$$

where Wy is the spatial lag variable, $\rho$ is a spatial autoregressive coefficient, X is the independent regression variable, $\beta$ is the regression coefficients, and $\varepsilon$ is the error term vector.

Conversely, the spatial error model considers the spatial impacts that are not wholly defined by the factors within the error terms and is presented in Equation (10).

$$y = \beta X + \lambda W \mu + \varepsilon \tag{10}$$

where W$\mu$ is a vector of spatially lagged errors, and $\lambda$ is a coefficient of spatial autoregressive

The study implemented the Lagrange Multiplier statistics to match the applied modeling methods for each year. The standard coefficients (beta weights) were used to examine the relative significance of composition and configuration metrics on estimating LST [43,45]. The regression model was run using GEODA version 1.8.16 [87].

## 2.7. Summary of Experimental Steps

Figure 5 illustrates three series of tasks that were conducted to analyze the spatial patterns of UGS and LST. These are (1) classification of the LULC and validation process; (2) LST analysis; and (3) examining the effects of UGS patterns on the LST.

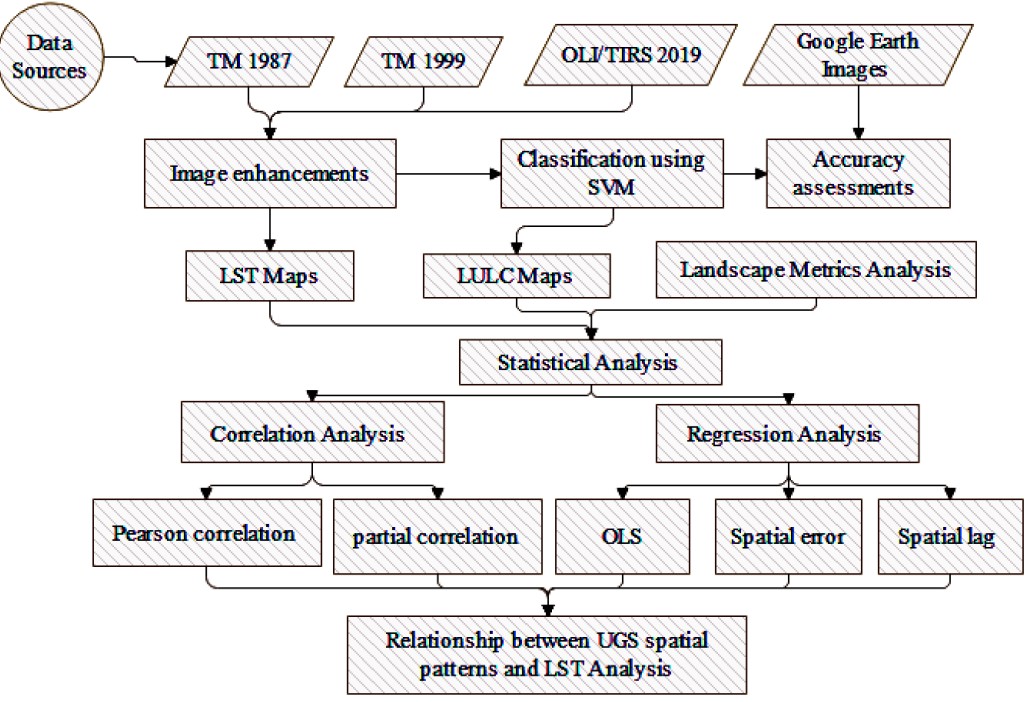

**Figure 5.** Summary of experimental steps in UGS patterns and LST relationship analysis.

## 3. Results

### 3.1. LULC Changes in Addis Ababa Metropolitan Regions

Figure 6 shows the changes in the land use pattern of the Addis Ababa metropolitan regions from the 1987 to 2019 study period. The area of non-urban areas (vegetation, grass, agriculture, and water) has rapidly been reduced across the study periods. In contrast, the area of the urban region has significantly increased over the past 32 years (Figure 6). The overall and kappa coefficients of LULC classes were 87.8, 88.2, and 89.4 in 1987, 1999, and 2019, respectively. The kappa values were 0.85 for 1987 and 1999, respectively, and 0.88 for 2019. These values indicated that the generated data had responsibly high accuracy.

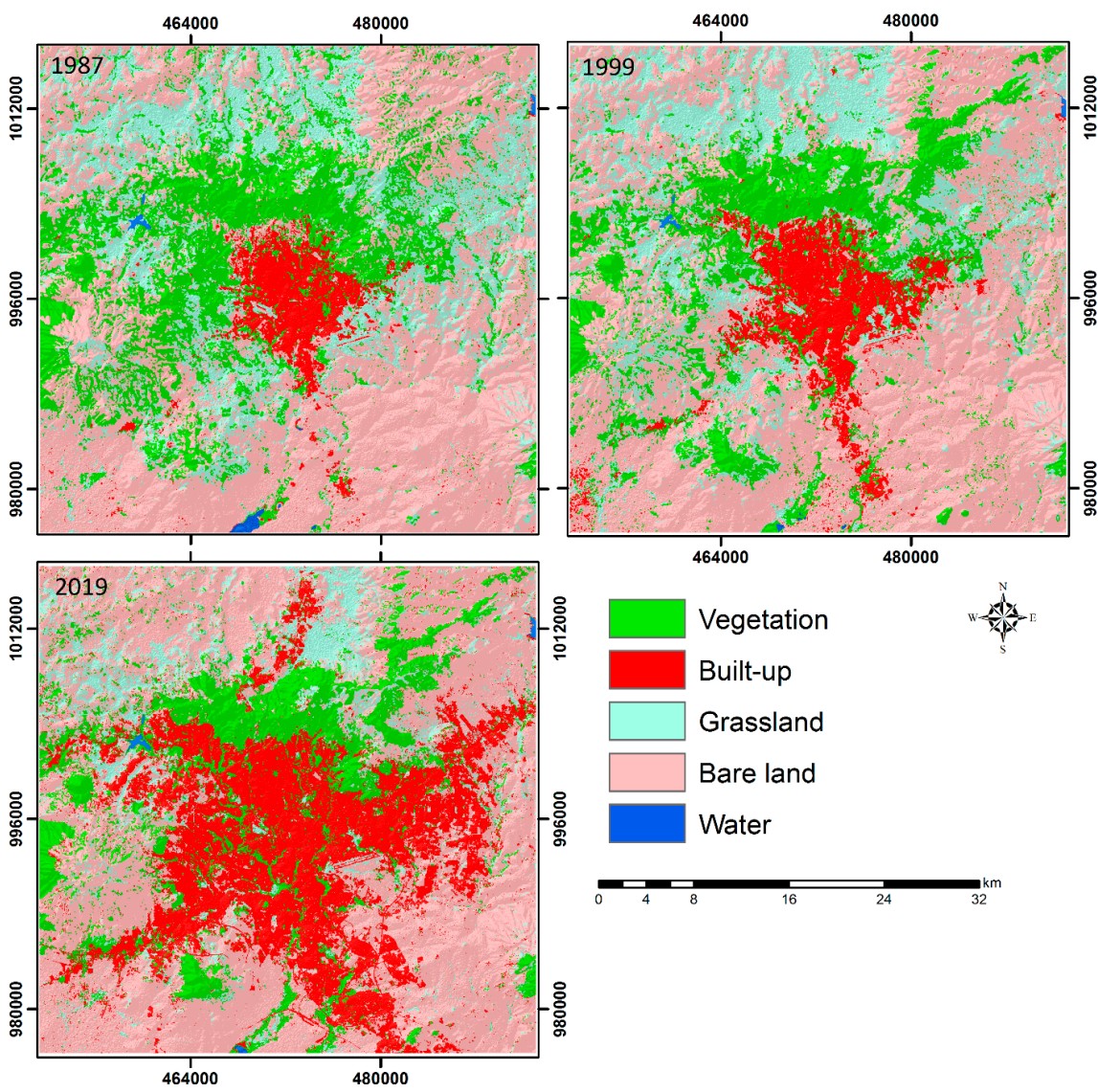

**Figure 6.** LULC Maps of Addis Ababa metropolitan area.

### 3.2. Space and Time-Based Pattern of UGS in the Addis Ababa Metropolitan Area

Table 3 shows the relative change patterns of UGS and the corresponding landscape metrics. The proportion of urban green space (PGS) in Addis Ababa metropolitan decreased gradually from 44.3% in 1987 to 33.4% in 1999, followed by a considerable decrease to 26.3% in 2019 (Table 3). In terms of fragmentation and aggregation, the PD values of the UGS gradually increased between 1987 and 1999, and then sharply increased further in 2019. The result indicated that the UGS was disaggregated

across the study periods, but UGS patch fragmentation was more significant in 2019, as evidenced by a sharp rise in the PD values: 7 in 1999 to 27 in 2019. The shape complexity of UGS, which was calculated by ED, revealed that changes in the overall complexity of the patches became more complex and irregular patterns from 1987 to 2019. This trend was high during the 1999–2019 periods, as evidenced by a rapid increase in the ED values, from 111.4 m/ha in 1999 to 2015.1 m/ha in 2019 (Table 3). Whereas, the connectivity among UGS patches increased notably from 1987 to 2019 and was generally increased by 41 m during this study period (from 42 m of MNN in 1987 to 91.36 m in 2019) (Table 3).

**Table 3.** Landscape metrics results from 1987 to 2019.

| Metrics | Unit | 1987 | 1999 | 2019 |
|---------|------|------|------|------|
| PGS | % | 44.33 | 38.36 | 26.31 |
| PD | Number/100 ha | 5 | 7 | 27 |
| ED | m/ha | 96.30 | 111.39 | 215.14 |
| MNN | m | 42.00 | 57.88 | 91.36 |

In general, it was inferred that Addis Ababa lost its green space from 1987 to 2019, at about 12% between 1999 and 2019. Concerning the spatial patterns, UGS patches have become smaller, more complex in shape, and more fragmented overall.

### 3.3. Effects of Spatial Patterns of UGS on LST

Across the study periods, the Pearson correlation analysis revealed that most metrics were significant at all analytical units, except MNN at large spatial scales (Table 4). Composition metrics (PGS) showed a negative correlation with LST, suggesting that the more UGS patches give a higher cooling effect. Among the configuration metrics, PD revealed a positive correlation with LST, indicating that the low fragmented UGS patches show a more effective cooling signature. In contrast, the ED had a negative correlation with LST across all analytical units. In contrast, MNN revealed a positive, significant correlation with LST only at analytical units of 120 m, 240 m, and 360 m. Similar to PGS, the intensity of the correlations between the three configuration metrics (PD, ED, and MNN) and LST also generally decreased with the increase of the analysis scale.

After controlling for the impact of PGS, the correlations (i.e., partial correlations) between LST and configuration metrics were significantly changed and are shown in Table 4. From the table, four features were notable: (i) the intensity of the partial correlations, calculated by the correlation coefficients significantly declined relative to their equivalent Pearson correlation coefficients. (ii) Some configuration metrics and LST became insignificant (e.g., MNN). (iii) Significantly correlated at only smaller scales (e.g., ED was significantly correlated to LST at less than or equal to 360 m, and reduced as the spatial magnitude increased), and (iv) More importantly, the association between some configuration metrics and LST shifted their directions. For instance, the correlation between MNN and LST were shifted from positive to negative when the analytical unit was between 360 and 960 m, 360 to 840 m, and 360 to 840 m in the year 1987, 1999, and 2019, respectively. In addition, the correlation of ED with LST was shifted from negative to positive across all analytical units except at 1080 m in the year 1987.

**Table 4.** Correlation coefficients between Landscape metrics and LST. The bold rows indicate the partial correlation analysis, where for proportion of urban green space (PGS), the control variables were the configuration metrics, and for configuration metrics, the control variable was PGS.

| Year | Variable | Scale | | | | | | | | |
|---|---|---|---|---|---|---|---|---|---|---|
| | | 120 m | 240 m | 360 m | 480 m | 600 m | 720 m | 840 m | 960 m | 1080 m |
| 1987 | PGS | −0.658 ** | −0.647 ** | −0.661 ** | −0.610 ** | −0.529 ** | −0.509 ** | −0.449 ** | −0.427 ** | −0.470 ** |
| | | **−0.465 **** | **−0.347 **** | **−0.305 **** | **−0.300 **** | **−0.298 **** | **−0.146 *** | **−0.103** | **−0.179 *** | **−0.175 *** |
| | PD | 0.570 ** | 0.625 ** | 0.631 ** | 0.598 ** | 0.503 ** | 0.552 ** | 0.467 ** | 0.375 ** | 0.429 ** |
| | | **0.259 **** | **0.282 **** | **0.230 **** | **0.267 **** | **0.205 **** | **0.277 **** | **0.219 **** | **0.019** | **0.001** |
| | ED | −0.484 ** | −0.536 ** | −0.586 ** | −0.513 ** | −0.417 ** | −0.447 ** | −0.418 ** | −0.380 ** | −0.410 ** |
| | | **0.209 **** | **0.161 *** | **0.077** | **0.166 *** | **0.192 *** | **0.089** | **0.077** | **0.032** | **−0.018** |
| | MNN | 0.392 ** | 0.350 ** | 0.194 ** | 0.038 | 0.119 | 0.087 | 0.016 | 0.020 | 0.050 |
| | | **0.103** | **0.105** | **−0.008** | **−0.076** | **−0.078** | **−0.028** | **−0.135** | **−0.022** | **0.052** |
| 1999 | PGS | −0.649 ** | −0.686 ** | −0.606 ** | −0.566 ** | −0.514 ** | −0.555 ** | −0.530 ** | −0.468 ** | −0.420 ** |
| | | **−0.483 **** | **−0.468 **** | **−0.376** | **−0.337 **** | **−0.305 **** | **−0.394 **** | **−0.235 **** | **−0.289 **** | **−0.279 **** |
| | PD | 0.548 ** | 0.581 ** | 0.519 ** | 0.482 ** | 0.453 ** | 0.449 ** | 0.501 ** | 0.406 ** | 0.352 ** |
| | | **0.282 **** | **0.214 **** | **0.148** | **0.129** | **0.113 *** | **0.071** | **0.131** | **0.021** | **−0.002** |
| | ED | −0.414 ** | −0.492 ** | −0.466 ** | −0.457 ** | −0.397 ** | −0.405 ** | −0.450 ** | −0.346 ** | −0.325 ** |
| | | **0.194 **** | **0.183 **** | **0.126** | **0.112 *** | **0.143 *** | **0.188 *** | **0.104** | **0.108** | **0.133** |
| | MNN | 0.262 ** | 0.398 ** | 0.142 * | 0.111 * | 0.146 * | 0.055 | 0.067 | 0.104 | −0.004 |
| | | **0.023** | **0.144 *** | **−0.045** | **−0.079** | **−0.001** | **−0.102** | **−0.042** | **0.006** | **−0.042** |
| 2019 | PGS | −0.797 ** | −0.823 ** | −0.769 ** | −0.708 ** | −0.733 ** | −0.652 ** | −0.613 ** | −0.551 ** | −0.490 ** |
| | | **−0.613 **** | **−0.580 **** | **−0.487 **** | **−0.410 **** | **−0.400 **** | **−0.246 **** | **−0.249 **** | **−0.253 **** | **−0.098** |
| | PD | 0.740 ** | 0.708 ** | 0.649 ** | 0.612 ** | 0.649 ** | 0.598 ** | 0.580 ** | 0.504 ** | 0.494 ** |
| | | **0.466 **** | **0.245 **** | **0.172 *** | **0.177 *** | **0.195 *** | **0.122 *** | **0.195 *** | **0.058** | **0.111** |
| | ED | −0.623 ** | −0.673 ** | −0.645 ** | −0.587 ** | −0.624 ** | −0.550 ** | −0.562 ** | −0.477 ** | −0.436 ** |
| | | **0.118 *** | **0.053** | **0.002** | **0.037** | **0.054** | **0.065** | **0.126** | **0.045** | **−0.004** |
| | MNN | 0.294 ** | 0.308 ** | 0.139 ** | 0.209 ** | 0.120 * | 0.098 | −0.069 | 0.054 | 0.053 |
| | | **0.026** | **0.050** | **−0.023** | **−0.030** | **−0.018** | **−0.064** | **−0.019** | **0.022** | **0.027** |

$** p < 0.01, * p < 0.05.$

### 3.4. Influences of UGS Spatial Size Pattern on LST

The correlation between the implemented landscape metrics and LST appears to be regularly decreasing with spatial size (Table 4). Whereas, some configuration metrics can influence on LST separately of the composition metrics at smaller scales. For example, the correlation of PD continued to be significant with LST at analytical units between 120 m to 600 m. This tendency is also noted when considering the influence of PGS across the study periods. The results imply that the patch density has a more prominent effect on LST than the other configuration metrics at smaller scales. It is noted that the optimum spatial scale of UGS for exploring the relationship between LST and spatial metrics could be between 120–600 m.

Indeed, this study did not find nor does it seem feasible to precisely identify a scale at which the correlation between LST and metrics converge. However, most metrics were generally more predictable at 240 m. For instance, after controlling for the impact of composition metrics (PGS), connectivity between the green spaces revealed a significant positive effect on LST only on the spatial scale of 240 m, as evidenced by MNN values in Table 4. Thus, 240 m × 240 m was applied as the optimum size for further analysis.

### 3.5. The Relative Significance of Composition and Configuration of UGS on LST

Table 5 shows that the results from spatial error, Lag, and OLS linear regression models, the PGS had significant negative impacts on LST, across all the study periods. Additionally, PGS was the most significant indicator of LST, playing a considerably more significant role in forecasting LST relative to the other spatial configuration factors (Table 5). Among the three configuration metrics, PD played a much more significant and positive effect in estimating LST (Table 5). Other metrics, ED and MNN had positively significant effects on LST in the years 1985 and 1999.

**Table 5.** Results for regression modeling. The Z-value shows the relative implication of each variable in the model. R$^2$ (coefficient of determination) and AIC (Akaike info criterion) reveal the general enactment of each model.

| Year | Model | Variable | Coefficient | R2 | Z-Value | AIC |
|---|---|---|---|---|---|---|
| 1987 | Spatial error | PGS | −2.814 ** | 0.482 | −4.620 | 831.739 |
| | | ED | 3.589 ** | | 2.833 | |
| | | PD | 0.203 ** | | 3.809 | |
| | | MNN | 0.001 * | | 2.259 | |
| | Spatial LAG | PGS | −2.822 ** | 0.480 | −4.572 | 831.379 |
| | | ED | 3.409 ** | | 2.833 | |
| | | PD | 0.217 ** | | 4.061 | |
| | | MNN | 0.001 * | | 2.172 | |
| | OLS | PGS | −3.197 ** | 0.463 | −5.157 | 835.587 |
| | | ED | 3.252 ** | | 2.499 | |
| | | PD | 0.196 ** | | 3.565 | |
| | | MNN | 0.001 * | | 2.054 | |
| 1999 | Spatial error | PGS | −4.226 ** | 0.584 | −8.091 | 829.129 |
| | | ED | 4.654 ** | | 3.929 | |
| | | PD | 0.152 ** | | 3.783 | |
| | | MNN | 0.001 | | 1.372 | |
| | Spatial LAG | PGS | −3.916 ** | 0.565 | −7.432 | 836.386 |
| | | ED | 4.187 ** | | 3.558 | |
| | | PD | 0.145 ** | | 3.419 | |
| | | MNN | 0.001 | | 1.564 | |
| | OLS | PGS | −0.188 ** | 0.507 | −7.494 | 854.861 |
| | | ED | 3.131 * | | 2.471 | |
| | | PD | 0.125 ** | | 2.726 | |
| | | MNN | 0.001 | | 1.627 | |
| 2019 | Spatial error | PGS | −4.517 ** | 0.661 | −7.571 | 530.529 |
| | | PD | 0.121 ** | | 2.914 | |
| | | ED | 0.442 | | 0.472 | |
| | | MNN | 0.0003 | | 0.7335 | |
| | Spatial LAG | PGS | −4.481 ** | 0.655 | −7.460 | 531.278 |
| | | PD | 0.121 ** | | 2.928 | |
| | | ED | 0.640 | | 0.667 | |
| | | MNN | 0.0002 | | 0.5667 | |
| | OLS | PGS | −4.468 * | 0.634 | −7.081 | 536.774 |
| | | PD | 0.118 ** | | 2.705 | |
| | | ED | 0.201 | | 0.203 | |
| | | MNN | 0.0002 | | 0.6108 | |

** $p < 0.01$, * $p < 0.05$.

The results indicated that composition, fragmentation, and shape complexity characteristics of UGS spatial patterns affected LST. Moreover, the results showed that the spatial error model outperformed its correspondent Lag, and OLS models, for each of the periods. This was evidenced by the higher R$^2$ values and the lower AIC of the spatial error model compared to the other two models (Table 5).

## 4. Discussion

The relative importance of both spatial composition and configuration of UGS on LST was almost similar across the study periods. PGS was the critical factor in estimating LST. This is consistent with prior studies that suggested that the amount of green space plays a more important role in the cooling effects of the city than their configurations [42,45,46,85]. The overall results suggest that the fraction of

green patches, their geometrics shape, degree of aggregation and fragmentation, and the proximity distance of patches to each other, are the main determinants of LST in urban regions. In other words, for the greater cooling effect, more patches of UGS distribution and a reasonable spatial configuration are important considerations to impact LST and reduce the urban heat.

### 4.1. Scale-Dependence of UGS Spatial Patterns and LST Relationships

The association between PGS and LST was negative at all spatial scales, across the study periods. This is consistent with the past results that showed that the amount of vegetation cover is a significant feature in mitigating UHI effects [32,45,53]. The study also found that increasing the spatial scale, the relationship between landscape metrics and LST, including both composition and configuration, becomes weaker, highlighting a scale effect. The results are consistent with the recent studies found elsewhere [46]. Furthermore, a nonlinear relation is noted between LST and vegetation cover- the LST reduction by enhancing the amount of vegetation cover was relatively lower at larger scales (Table 6). This is likely due to higher diversity in green features of land covers and complex combinations typically occurring with the higher green cover than for small scales. Hence, smaller green fractions are likely less sensitive to the effect of changes in the proportion of vegetation cover on LST.

**Table 6.** Results from OLS linear regression. The effects of UGS on LST was predicted by PGS.

| Year | | Scale | | | | | | | | |
|------|------|-------|-------|-------|-------|-------|-------|-------|-------|-------|
| | | 120 m | 240 m | 360 m | 480 m | 600 m | 720 m | 840 m | 960 m | 1080 m |
| 1987 | Coef. | −0.894 ** | −0.508 ** | −0.507 ** | −0.573 ** | −0.605 ** | −0.307 * | −0.230 * | −0.469 * | −0.428 * |
| | $R^2$ | 0.486 | 0.475 | 0.467 | 0.419 | 0.303 | 0.320 | 0.241 | 0.184 | 0.225 |
| 1999 | Coef. | −0.602 ** | −0.633 ** | −0.621 ** | −5.133 ** | −4.158 ** | −4.158 ** | −2.996 ** | −0.660 ** | −0.728 ** |
| | $R^2$ | 0.470 | 0.510 | 0.384 | 0.335 | 0.286 | 0.333 | 0.298 | 0.237 | 0.195 |
| 2019 | Coef. | −0.585 ** | −0.219 ** | −0.640 ** | −0.599 ** | −0.626 ** | −0.587 ** | −0.647 ** | −0.507 * | −0.268 * |
| | $R^2$ | 0.726 | 0.707 | 0.606 | 0.518 | 0.556 | 0.441 | 0.401 | 0.312 | 0.258 |

$** p < 0.01, * p < 0.05$.

As noted, while the study did not find the exact spatial scale at which the greatest correlation has occurred, but it appeared that most metrics became more predictable at 240 m. This could be a robust feature considering the following three reasons, as demonstrated in this study. Primarily, after controlling for the effects of PGS, patch density of UGS revealed significant positive impacts on LST at the spatial extents between 120 m and 600 m in each studied period (see Table 4). Secondly, after controlling for the impact of composition metrics, connectivity between the green spaces revealed a significant positive effect on LST only on the spatial scale of 240 m in the year 1999, as evidenced by MNN values in Table 4. Third, such spatial scales were also suggested in earlier studies [28,43,57,58], though varied data used and approaches applied. Together this suggests that a spatial scale around 240 m could be an optimal spatial size to investigate the connection between UGS spatial patterns and LST.

The study results also revealed that spatial autocorrelation affected the association between LST and landscape metrics. Results from the spatial error and spatial lag models were higher than OLS in terms of $R^2$, regression coefficients, and AIC values. This indicated that temperature in a specific region was affected by and affected the temperature in its surrounding samples (via advection). A similar finding was also suggested by [45,85]. The findings indicated that the LST at a location was not only dependent on the spatial arrangement of UGS in that region but also broader distribution in nearby areas.

### 4.2. The Spatial Composition and Configuration of UGS Effects on LST

While the amount of urban green space was the critical reason for cooling urban LST, urban greens arrangement was also associated with LST distributions. Arrangement of green space, including patch density, patches connectivity, shape complexity, and edge density, revealed positive effects on LST.

These spatial configurations of urban green spaces' effects on LST; however, varied with the earlier studies in terms of significance, direction, and magnitude. The positive association between PD and LST in this study, for instance, is consistent with studies [42,53,85], suggesting aggregated greenspace is better than a fragmented greenspace in reducing LST, while others reported the opposite [16,49,51]. This study found that all analytical units revealed a positive correlation of the green space patch density with LST. In addition, after controlling for the effects of a green space percent, the PD was positively significant to LST within 120–600 m across the study periods. Similarly, the contradictory results of UGS configuration effects on the LST in different cities were documented by several studies. Edge density of UGS, for instance, was found to be negatively associated with LST in Kansas, USA [42], Baltimore, USA [45], and Berlin, Germany [52], but positive in Singapore [53]. Rising of the edge density may potentially result in a rise of shade offered by vegetation to adjoining surfaces [51,85]. It may also increase energy flow and coupling between vegetation and their neighboring areas [56]. Thereby, taking into account only the process of shading, rising edge density will result in lower LST. Conversely, high edge density is mostly a result of more disaggregated UGS. As small and dispersed UGS has a higher temperature than that of aggregated and larger patches [43,53], which can decrease the effectiveness of surface albedo and evapotranspiration. This study also highlighted that the edge density of UGS had positive effects on LST across the study periods, suggesting that the complex shape and irregular patches of green spaces, the higher in LST. In other words, the UGS with simpler and regular patches leads to reduced LST.

The inconsistent results of configuration metrics reported by different researchers are likely due to the amount of UGS considered in the studies. For instance, in 1987, the amount of UGS was 44.3%. This figure reduced to 38.4% in 1999 and finally reached 26.3% in 2019 (see Table 3). At the same time, the effects of configuration metrics were changed in terms of direction, significance, and magnitude. For instance, the average nearest distances among each patch of UGS had a positive impact on LST in 1987 (see Table 4). However, in the case of data for 2019, it revealed insignificant, even changed its direction to negative at larger scales (see Table 4). This suggests that the amounts of green spaces can decide the effectiveness of configuration metrics to predict the relationship between LST and UGS patterns. Simultaneously, high fragmentation, complex shape pattern, and decreasing the connectivity among each patch observed in 2019 than the other periods. This may suggest that when patches are highly complicated and disconnected, the configuration's role could be declined. These outcomes can improve the understanding of the irregularity impacts of the spatial configuration of UGS on LST from earlier studies.

Furthermore, the irregularity impacts of configuration metrics on LST could be due to the spatial scale effects. The effects of configuration metrics were variable across the study periods. The average nearest distance between patches (MNN) was, for instance, notable in scale dependence with LST, where the relationships of MNN shifted from positive at a smaller spatial size to negative at the larger spatial size (see Table 4). That is, the correlation between configuration patterns of UGS and LST was sensitive to the spatial scales. The multi-scale analysis can assist resolve these inconsistencies. By exploring the relationship between UGS and LST pattern metrics under various analytical units, the insight into the impacts of UGS patterns on LST can be enhanced.

In general, considering the significant impacts of spatial size [16,28,43,45], the satellite image resolution [42,55], and the choice of statistical approach [51,57], comparing the UGS and LST relations across the studies could be challenging. This multi-scale and multi-year analysis with medium-resolution thermal data assists the interpretation and extension of prior that considered a single snapshot data in multiple cities, e.g., [39,45]. This could have the potential to identify the spatial composition and configuration relationships with LST due to its multiple periods in the same area to confirm its consistency.

### 4.3. Methodological Implications for Urban Greenspace Spatial Pattern Analysis

The study findings emphasize the need for controlling the influences of the percentage of green spaces when analyzing the spatial arrangement of UGS impacts on LST. For each studied period, after controlling for the influences of PGS, the association between LST and configuration metrics radically shifted, related to outcomes from the Pearson correlation analysis. The correlation between edge density and LTS, for instance, changed from negative to positive at all scales across the study periods. Additionally, the correlation between LTS and average nearest neighbor patch distance has shifted from positive to negative after or equal to 480 m. This is due to most of the configuration factors naturally correlated to PGS (Table A1; [45,85]). For instance, edge density had a significant negative connection with LST as per Pearson correlation analysis (r = −0.54, −0.49, and −0.67, *p* < 0.01 Table 3) in 1987, 1999, and 2019, respectively at the scale of 240 m. The correlation was due to the highly significant positive association between edge density and PGS (r = 0.83, 0.77, and 0.74, *p* < 0.01 Table A1) in 1987, 1999, and 2019, respectively. After controlling for the influences of PGS, edge density indeed had a significant positive connection with LST. Thus, it is important to apply a statistical approach such as multiple regression models and partial correlation, instead of Pearson correlation, to assess the relative contributions configuration and percent cover of green spaces to the LST. Applying Pearson correlation analysis alone could produce distorted results.

This research also suggested that the spatial pattern of UGS in Addis Ababa can be efficiently computed using only two metrics: PD and PGS, which represent configuration and composition features of spatial patterns. Each of these two metrics is the most important metrics related to their matching element, as distinguished by OLS, Lag, and Spatial error models. These metrics could be steadily applied to describe the temporal dynamics in LST distributions. These findings can help by minimizing the need to measure the redundant metrics as well as interpreting multiple metrics.

Furthermore, statistical methods, such as structural equation modeling and patch analysis, have been increasingly applied to detect the complexity and nested associations among social situations, land cover, landscape patterns, and surface temperatures [42,88]. Such assessments potentially assist in detecting the direct or indirect influences of the spatial structure of UGS on LST. Among the demonstrated spatial regression approaches, the spatial error model was more convenient than OLS in predicting LST, based on the pattern of UGS, as evidenced by their regression coefficient, $R^2$, and AIC, across the study periods. This is especially relevant considering that the spatial patterns of UGS and LST are more likely to invalidate the assumption of independence among each patch, which did not occur in the frequently used statistical approaches such as OLS and Pearson correlation analysis. This study, therefore, again highlights that for spatial datasets, including the exploration of UGS spatial pattern effects on LST, the spatial regression approaches offer more robust evidence by incorporating spatial autocorrelation into the modeling to overcome the spatial clustering error from spatial dependence. These findings thus underlined the results of earlier studies [43,85] that the importance of the spatial regression model can overcome the concerns of spatial autocorrelation when exploring LST-UGS pattern relationships, which cannot be considered by OLS.

## 5. Conclusions

Urban green space has considerable cooling potential in mitigating urban heat. It is well-understood that increasing the amount of UGS can reduce the temperature of the city, but studies on the effects of UGS configuration have revealed inconsistent results. This study conducted multi-temporal, multiple statistical assessments, and varied scales to examine the connection between LST and green space patterns in Addis Ababa metropolitan area and further investigate the reasons for the inconsistency in prior studies. Landsat images were used to generate the LST and land cover maps. The main findings are summarized here.

(1) The UGS patches with aggregated, regular and simple shape and connectivity across the urban landscape are more effective in decreasing the LST than that of fragmented and complicated shape patterns. Thus, in addition to simply increasing the amount of UGS, optimizing the spatial structure of UGS, can be an effective and useful action to mitigate the UHI impacts. (2) The study also confirmed that the spatial pattern of UGS (in Addis Ababa) could be efficiently computed using only two metrics: PGS and PD, which represent, composition and configuration characteristics of spatial patterns. (3) Changing the spatial size had a considerable influence on the association between LST and UGS patterns. As the spatial scale increased, the influences of landscape metrics on LST, including both composition and configuration, become weaker, indicating spatial scale effects. (4) It is also noted that the configuration metrics (PD, ED, and MNN) are more sensitive to spatial scale than composition metrics (PGS). (5) The relationships between configuration metrics and LST could be changed when applying different statistical methods. This result underlines the importance of controlling the effects of PGS when calculating the impacts of the spatial configuration of UGS on LST. These outcomes provide additional understanding of the varying findings from earlier studies, likely confounded by the different approaches used (e.g., Partial correlation analysis versus Pearson correlation). (6) The relationships between LST and landscape metrics could be affected by spatial autocorrelation. In general, the study highlighted that applying different statistical approaches, spatial scale, and abundance of UGS can appropriately determine the effectiveness of configuration metrics to predict the association between LST and UGS patterns.

Urban heat island reduction is a potential mitigative strategy to decrease heat stress. Urban planning can increase the resilience of urban regions through enhancements to the green infrastructure and built environments. The findings of this study have potential implications for urban planners, suggesting more green space distribution with careful consideration of spatial configuration is important for effectively achieving the reduction in urban heating. Green infrastructure planning needs to consider both composition and configuration analyses for scenario planning. However, this study did not calculate the intensity of UHI, and future studies can incorporate it.

**Author Contributions:** Conceptualization, B.K.T. and N.C.; methodology, B.K.T.; software, B.K.T.; formal analysis, B.K.T.; investigation, B.K.T.; validation, B.K.T., N.C., and X.Z.; resources, N.C.; B.K.T.; data curation, writing—original draft preparation, B.K.T.; writing—review and editing, B.K.T., N.C., X.Z., and D.N.; visualization, B.K.T., N.C., and X.Z.; supervision, X.Z and N.C.; project administration, N.C.; funding acquisition, X.Z. and N.C. All authors have read and agreed to the published version of the manuscript.

**Funding:** This research was supported in part by the National Key R&D Program (No. 2018YFB2100500), National Natural Science Foundation of China program (No. 41890822, 41801339), Creative Research Groups of Natural Science Foundation of Hubei Province of China (No. 2016CFA003) and U.S. NSF OAC-1835739.

**Acknowledgments:** The authors would like to express respect and appreciation to the editors and reviewers for their professional comments and suggestions.

**Conflicts of Interest:** The authors declare no conflict of interest.

# Appendix A

Table A1. Expressive statistics of landscape metrics of UGS and LST.

| Scale | Variables | 1987 | | | | 1999 | | | | 2019 | | | |
|---|---|---|---|---|---|---|---|---|---|---|---|---|---|
| | | MNN | ED | PD | PGS | MNN | ED | PD | PGS | MNN | ED | PD | PGS |
| 120 m | MNN | 1 | | | | 1 | | | | 1 | | | |
| | ED | −0.488 ** | 1 | | | −0.452 ** | 1 | | | −0.372 ** | 1 | | |
| | PD | 0.503 ** | −0.738 ** | 1 | | 0.317 ** | −0.686 ** | 1 | | 0.381 ** | −0.734 ** | 1 | |
| | PGS | −0.458 ** | 0.788 ** | −0.683 ** | 1 | −0.413 ** | 0.703 ** | −0.647 ** | 1 | −0.261 ** | 0.689 ** | −0.647 ** | 1 |
| 240 m | MNN | 1 | | | | 1 | | | | 1 | | | |
| | ED | −0.385 ** | 1 | | | −0.421 ** | 1 | | | −0.265 ** | 1 | | |
| | PD | 0.402 ** | −0.824 ** | 1 | | 0.401 ** | −0.743 ** | 1 | | 0.335 ** | −0.727 ** | 1 | |
| | PGS | −0.394 ** | 0.830 ** | −0.780 ** | 1 | −0.443 ** | 0.773 ** | −0.724 ** | 1 | −0.311 ** | 0.739 ** | −0.716 ** | 1 |
| 360 m | MNN | 1 | | | | 1 | | | | 1 | | | |
| | ED | −0.177 ** | 1 | | | −0.103 | 1 | | | 0.001 | 1 | | |
| | PD | 0.226 ** | −0.837 ** | 1 | | 0.158 ** | −0.779 ** | 1 | | 0.135 ** | −0.762 ** | 1 | |
| | PGS | −0.293 ** | 0.866 ** | −0.808 ** | 1 | −0.270 ** | 0.818 ** | −0.765 ** | 1 | −0.204 ** | 0.795 ** | −0.737 ** | 1 |
| 480 m | MNN | 1 | | | | 1 | | | | 1 | | | |
| | ED | 0.156 * | 1 | | | −0.048 | 1 | | | −0.074 | 1 | | |
| | PD | −0.005 | −0.824 ** | 1 | | 0.296 ** | −0.775 ** | 1 | | 0.335 ** | −0.731 ** | 1 | |
| | PGS | −0.098 * | 0.868 ** | −0.813 ** | 1 | −0.228 ** | 0.848 | −0.774 ** | 1 | −0.277 ** | 0.811 ** | −0.747 ** | 1 |
| 600 m | MNN | 1 | | | | 1 | | | | 1 | | | |
| | ED | 0.022 * | 1 | | | 0.052 * | | | | 0.140 * | 1 | | |
| | PD | 0.237 ** | −0.820 ** | 1 | | 0.198 ** | −0.835 ** | | | 0.218 ** | −0.747 ** | 1 | |
| | PGS | −0.169 * | 0.877 ** | −0.829 ** | 1 | −0.147 | .0874 ** | −0.827 ** | 1 | −0.115 | 0.848 ** | −0.768 ** | 1 |
| 720 m | MNN | 1 | | | | 1 | | | | 1 | | | |
| | ED | 0.169 * | 1 | | | 0.173 ** | 1 | | | 0.125 | 1 | | |
| | PD | 0.146 * | −0.795 ** | 1 | | 0.207 ** | −0.776 ** | 1 | | 0.318 ** | −0.754 ** | 1 | |
| | PGS | −0.047 | 0.889 ** | −0.822 ** | 1 | −0.090 | 0.853 ** | −0.819 ** | 1 | −0.141 * | 0.859 ** | −0.832 ** | 1 |

**Table A1.** *Cont.*

| Scale | Variables | 1987 | | | | 1999 | | | | 2019 | | | |
|-------|-----------|------|------|------|------|------|------|------|------|------|------|------|------|
| | | MNN | ED | PD | PGS | MNN | ED | PD | PGS | MNN | ED | PD | PGS |
| 840 m | MNN | 1 | | | | 1 | | | | 1 | | | |
| | ED | 0.074 | 1 | | | 0.174 * | 1 | | | 0.316 ** | 1 | | |
| | PD | 0.279 ** | −0.828 ** | 1 | | 0.217 ** | −0.808 ** | 1 | | −0.332 ** | −0.907 ** | 1 | |
| | PGS | −0.104 | 0.893 ** | −0.858 ** | 1 | −0.029 | 0.900 ** | −0.852 ** | 1 | 0.071 | 0.922 ** | −0.841 ** | 1 |
| 960 m | MNN | 1 | | | | 1 | | | | 1 | | | |
| | ED | 0.146 * | 1 | | | 0.215 ** | 1 | | | 0.226 ** | 1 | | |
| | PD | 0.252 ** | −0.820 ** | 1 | | 0.264 ** | −0.794 ** | 1 | | 0.166 * | −0.851 ** | 1 | |
| | PGS | −0.050 | 0.909 ** | −0.872 ** | 1 | −0.039 | 0.874 ** | −0.859 ** | 1 | 0.041 | 0.899 ** | −0.859 ** | 1 |
| 1080 m | MNN | 1 | | | | 1 | | | | 1 | | | |
| | ED | 0.271 ** | 1 | | | 0.174 * | 1 | | | 0.317 ** | 1 | | |
| | PD | 0.203 * | −0.801 ** | 1 | | 0.122 * | −0.863 ** | 1 | | 0.161 * | −0.790 ** | 1 | |
| | PGS | 0.019 | 0.888 ** | −0.877 ** | 1 | 0.021 | 0.905 ** | −0.898 ** | 1 | 0.068 | 0.909 ** | −0.881 ** | 1 |

** $p < 0.01$, * $p < 0.05$.

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
