# Peer review of "Spatial Configuration and Extent Explains the Urban Heat Mitigation Potential due to Green Spaces: Analysis over Addis Ababa, Ethiopia"

_remotesensing, doi:10.3390/rs12182876_

Round 1

Reviewer 1 Report

The authors of this manuscript presented a study with the goal to determine the relationship between urban green space (UGS) and land surface temperature (LST). Study area is area of Addis Ababa city region, and LANDSAT images from year 1987, 1999 and 2019 are used. Different landscape metrics and statistical approaches are examined. The impact of spatial scale is also investigated. Another goal of this study is to resolve the inconsistency in findings of other studies.

The study presented in this paper involves detailed analysis and manages to reach set goals. Every part of the study is well explained and supported by references. The presentation is good, it is very well structured and tables and figures are adequate and of good quality. Style is very good and it is not difficult to read the manuscript. Each finding in the study is explained in scientific way (using numbers and scientific terms) and it is followed by ‘non-scientific’ explanation which is simplified but easier to understand even to readers that are not familiar with the topic or even remote sensing. This makes the paper more approachable to a wider audience and gives another quality to the manuscript.

I have no major comments and the corrections that should be done are more of technical nature.

Minor comments

Line 28: acronym UHI is not explained.

Line 41: citation is in superscript (in many other lines this occurs again, e.g. 43, 45, 48…)

Lines 46-47: ‘Changes in land use land cover…’. This is not clear. Maybe this should have been ‘Changes in land use and cover…’

Lines 358-360: Line numbers are ‘hidden’ in the table, and explanation of acronym AIC cannot be read.

Author Response

For Reviewer 1

Re: Revision of manuscript ID: Remote Sensing- 890019

Remote Sensing Journal

Thank you very much for the email which included three reviewers' comments and suggestions on our manuscript entitled "Spatial Configuration and Extent Explains the Urban Heat Mitigation Potential due to Green Spaces: Analysis over Addis Ababa, Ethiopia" (Manuscript ID 890019). On behalf of my co-authors, I would like to express my great appreciation to the editor and reviewers. We are very encouraged that all reviewers agree with the significance of our findings, and recommend the publication in the journal of Remote Sensing. We appreciate their constructive and valuable comments and suggestions. Hence, we have revised our manuscript accordingly. Now we would like to submit the revised manuscript for your consideration. Based on the comments and suggestions, we have revised our paper and indicated the changes in the main document with the revision model.

Reviewer 1

R1C1: The authors of this manuscript presented a study with the goal to determine the relationship between urban green space (UGS) and land surface temperature (LST). Study area is area of Addis Ababa city region, and LANDSAT images from year 1987, 1999 and 2019 are used. Different landscape metrics and statistical approaches are examined. The impact of spatial scale is also investigated. Another goal of this study is to resolve the inconsistency in findings of other studies.

The study presented in this paper involves detailed analysis and manages to reach set goals. Every part of the study is well explained and supported by references. The presentation is good, it is very well structured and tables and figures are adequate and of good quality. Style is very good and it is not difficult to read the manuscript. Each finding in the study is explained in scientific way (using numbers and scientific terms) and it is followed by ‘non-scientific’ explanation which is simplified but easier to understand even to readers that are not familiar with the topic or even remote sensing. This makes the paper more approachable to a wider audience and gives another quality to the manuscript.

I have no major comments and the corrections that should be done are more of technical nature.

Minor comments

Line 28: acronym UHI is not explained.

Response: The authors would like to thank the reviewer for their efforts in providing a detailed review of our study. We appreciate your valuable comments and suggestions. We have carefully worked on revising the paper building of your suggestions. As suggested by the reviewer, we have explained the acronym of UHI on line 29. The updated and corrected parts of the manuscript were indicated by red color lines throughout the article.

R1C2:Line 41: citation is in superscript (in many other lines, this occurs again, e.g. 43, 45, 48…)

Response: The authors appreciate and agree with the reviewer's comment. Following your comments, the correction has been made in the revised manuscript.

R1C3:Lines 46-47: ‘Changes in land use land cover…’. This is not clear. Maybe this should have been ‘Changes in land use and cover…

Response: Thank you for this valuable comment. Following your comments, the correction has been made. As suggested, “Changes in land use land cover” replaced by “Changes in land use and land cover” (line 51-52), as indicated in the revised manuscript.

R1C4:Lines 358-360: Line numbers are ‘hidden’ in the table, and explanation of acronym AIC cannot be read.

Response: Thank you for this valuable comment. The suggested correction has been made (inline 400-405), as shown in the modified manuscript.

Reviewer 2 Report

I think the authors need more effort to improve the paper by deeper classification the urban green space's structure into different classes and its change. 

What I can see from the manuscript is only one class "vegetation" I suggest that, it can be classified as shurb/grass or semi-shurb, urban tree (park or roadside) and private garden or even small agricultural areas in the urban may exit, ect. Then spatially investigate its change and how its associate with the LST.
The findings of this work are weak and in the abstract significant points have not been mentioned.

Author Response

For Reviewer 2

Re: Revision of manuscript ID: Remote Sensing- 890019

Remote Sensing Journal

Thank you very much for the email which included three reviewers' comments and suggestions on our manuscript entitled "Spatial Configuration and Extent Explains the Urban Heat Mitigation Potential due to Green Spaces: Analysis over Addis Ababa, Ethiopia" (Manuscript ID 890019). On behalf of my co-authors, I would like to express my great appreciation to the editor and reviewers. We are very encouraged that all reviewers agree with the significance of our findings, and recommend the publication in the journal of Remote Sensing. We appreciate their constructive and valuable comments and suggestions. Hence, we have revised our manuscript accordingly. Now we would like to submit the revised manuscript for your consideration. Based on the comments and suggestions, we have revised our paper and indicated the changes in the main document with the revision model.

Reviewer 2

R2C1:I think the authors need more effort to improve the paper by deeper classification the urban green space's structure into different classes and its change. 

What I can see from the manuscript is only one class "vegetation" I suggest that, it can be classified as shrub/grass or semi-shrub, urban tree (park or roadside) and private garden or even small agricultural areas in the urban may exit, etc. Then spatially investigate its change and how its associate with the LST.

Response: We would like to thank the positive feedback from the reviewer. The authors appreciate your valuable comments and suggestions. We have carefully worked on revising the paper building of your comments. Please refer to the following responses for more details.

This study mainly focused on the effects of spatial patterns of urban green space on land surface temperature to figure out the inconsistency reports of previous studies. To accomplish this aim, the urban land cover was classified into five classes: Built-up area, vegetation cover, grassland, waterbody, and bare land (line 164-169). To analyze the relationship between urban green space and land surface temperature, three land cover classes (vegetation cover, grassland, and water body) has been treated as urban green space (UGS) while the rests are excluded. Therefore, this study did not investigate the relationship between each of land cover classes with LST rather than considering vegetation, grass, and water body together as UGS. Even though each land cover classes has its own heat response values but vegetation, grass, and water body classes have relatively high cooling effects than bare land and built-up areas. Hence, they are considered as UGS. Thereby, the authors believed in considering the three land cover classes (i.e., vegetation, grass, and water body) as urban greens space is an integrated and direct approach to explore the relationship between each land caver classes and LST. We hope you will agree with us as well.

R2C2: The findings of this work are weak and in the abstract significant points have not been mentioned.

Response: Thank you for this valuable comment and suggestions.  After receiving your comments, we have improved the manuscript and made changes accordingly. We have further explained the significant points in the abstracts section. This revised part is indicated in line 25-40 of the modified manuscript.

Reviewer 3 Report

The manuscript entitled “Spatial Configuration and Extent Explain the Urban Heat Mitigation Potential due to Green Spaces: Analysis over Addis Ababa Ethiopia” provides a methodology to calculate the LST by Landsat images and the use of some indexes to visualize how the urban green space can contribute to mitigating the urban heat island effect. In general, the paper is well structured, with valuable results presented and a good description of the methods used.

Before considering suitable to publish the paper in the Remote Sensing, the authors must review grammar English, mostly in the Abstract and Introduction sections. After that, the authors should introduce a subsection called “Experiments” at the end of the Materials and Methods section, to describe the steps based on the methods previously described, maybe in a diagram format.

Another point that the authors must clarify is the LST calculation, basically described on lines 171 to 173. The sentences described are not clear.

Some doubts that the authors could consider for review: Figure 3 is a result based on the methodology previously described; the authors described many times the UHI, but its intensity is not calculated on the manuscript; the author should add a 2000’s decade map to consider every decadal urban expansion – there is a gap between 90’s and 2010s.

Author Response

For Reviewer 3

Re: Revision of manuscript ID: Remote Sensing- 890019

Remote Sensing Journal

Thank you very much for the email which included three reviewers' comments and suggestions on our manuscript entitled "Spatial Configuration and Extent Explains the Urban Heat Mitigation Potential due to Green Spaces: Analysis over Addis Ababa, Ethiopia" (Manuscript ID 890019). On behalf of my co-authors, I would like to express my great appreciation to the editor and reviewers. We are very encouraged that all reviewers agree with the significance of our findings, and recommend the publication in the Journal of Remote Sensing. We appreciate their constructive and valuable comments and suggestions. Hence, we have revised our manuscript accordingly. Now we would like to submit the revised manuscript for your consideration. Based on the comments and suggestions, we have revised our paper and indicated the changes in the main document with the revision model.

Reviewer 3

R3C1:The manuscript entitled “Spatial Configuration and Extent Explain the Urban Heat Mitigation Potential due to Green Spaces: Analysis over Addis Ababa Ethiopia” provides a methodology to calculate the LST by Landsat images and the use of some indexes to visualize how the urban green space can contribute to mitigating the urban heat island effect. In general, the paper is well structured, with valuable results presented and a good description of the methods used.

Before considering suitable to publish the paper in the Remote Sensing, the authors must review grammar English, mostly in the Abstract and Introduction sections. After that, the authors should introduce a subsection called “Experiments” at the end of the Materials and Methods section, to describe the steps based on the methods previously described, maybe in a diagram format.

Response: The authors would like to thank the reviewer for a careful and thorough reading of this article and for the valuable comments and constructive suggestions, which help to enhance the quality of this manuscript. We have carefully worked on revising the paper building off your comments. Please refer to the following responses for more details.

After received your suggestions, we have improved the manuscript and made changes accordingly. Grammar errors have been checked and corrected by English Grammarly and similarly, native English-speaking experts did the proofreading. The modified and corrected parts of the manuscript were indicated by red color texts throughout the article. Furthermore, as suggested by the reviewer,  we introduce a subsection called Summary of Experimental Steps, which briefly illustrates the experimental steps in diagrams forms as indicated (line 292-299).

R3C2:Another point that the authors must clarify is the LST calculation, described on lines 171 to 173. The sentences described are not clear.

 Response: Thank you for this very important suggestion. Accordingly, the suggested correction has been made by reorganized the structures of the sentences (inline 182-183) sections in the modified manuscript.  

R3C3:Some doubts that the authors could consider for review: Figure 3 is a result based on the methodology previously described; the authors described many times the UHI, but its intensity is not calculated on the manuscript; the author should add a 2000’s decade map to consider every decadal urban expansion – there is a gap between 90’s and 2010s.

Response: Thank you for your valuable suggestions and comments. This study was aimed to explore the effects of urban green space spatial patterns on land surface temperature and further to identify the inconsistency reasons for the previous studies' results. To achieve this objective, the study used three Landsat images (TM-1987 and 1999 as well as OLI-2019). The selection of these images was based on the availability of cloud-free images at similar months (line 134-18). Due to the lack of cloud-free images in 2009 or 2010 on similar months, the study considered only three study periods (1987, 1999, and 2019 years). Furthermore, the study did not aim to explore the intensity of UHI and applied multiple temporal images to explore the temporal effects of spatial patterns of urban green space on land surface temperature rather than calculating the intensity of UHI.  Therefore, the authors agreed as the estimation of UHI intensity could be considered in future studies. We hope the reviewer will agree with us as well.

This manuscript is a resubmission of an earlier submission. The following is a list of the peer review reports and author responses from that submission.